# Towards Vine Water Status Monitoring on a Large Scale Using Sentinel-2 Images

**Eve Laroche-Pinel** [1,2,3,*], **Sylvie Duthoit** [1], **Mohanad Albughdadi** [1], **Anne D. Costard** [1], **Jacques Rousseau** [4], **Véronique Chéret** [2,3] and **Harold Clenet** [2,3]

1  TerraNIS, 12 Avenue de l'Europe, F-31520 Ramonville Saint-Agne, France; sylvie.duthoit@terranis.fr (S.D.); mohanad.albughdadi@terranis.fr (M.A.); anne.costard@terranis.fr (A.D.C.)
2  Ecole d'Ingénieurs de Purpan, 75 voie du TOEC, F-31076 Toulouse, France; veronique.cheret@purpan.fr (V.C.); harold.clenet@purpan.fr (H.C.)
3  UMR DYNAFOR, INRAE, Université de Toulouse, F-31326 Castanet-Tolosan, France
4  Groupe—Institut Coopératif du vin, La Jasse de Maurin, F-34970 Montpellier, France; jrousseau@icv.fr
*  Correspondence: eve.laroche-pinel@terranis.fr; Tel.: +33-067-903-2721

**Abstract:** Wine growing needs to adapt to confront climate change. In fact, the lack of water becomes more and more important in many regions. Whereas vineyards have been located in dry areas for decades, so they need special resilient varieties and/or a sufficient water supply at key development stages in case of severe drought. With climate change and the decrease of water availability, some vineyard regions face difficulties because of unsuitable variety, wrong vine management or due to the limited water access. Decision support tools are therefore required to optimize water use or to adapt agronomic practices. This study aimed at monitoring vine water status at a large scale with Sentinel-2 images. The goal was to provide a solution that would give spatialized and temporal information throughout the season on the water status of the vines. For this purpose, thirty six plots were monitored in total over three years (2018, 2019 and 2020). Vine water status was measured with stem water potential in field measurements from pea size to ripening stage. Simultaneously, Sentinel-2 images were downloaded and processed to extract band reflectance values and compute vegetation indices. In our study, we tested five supervised regression machine learning algorithms to find possible relationships between stem water potential and data acquired from Sentinel-2 images (bands reflectance values and vegetation indices). Regression model using Red, NIR, Red-Edge and SWIR bands gave promising result to predict stem water potential ($R^2 = 0.40$, $RMSE = 0.26$).

**Keywords:** vineyard; water status; Sentinel-2; precision viticulture

## 1. Introduction

Water management is an important factor for wine growing especially in the Mediterranean regions. Vineyards, in these regions, require a sufficient amount of water in key development stages such as flowering, veraison and berry growth [1,2]. However, episodes of drought have significantly impacted Mediterranean vineyards leading to yield loss and irreversible damages over the last few years [3,4]. These constraints increase the considerations on the development of irrigation systems [5]. However, in numerous regions water restrictions are set up due to the lack of water or to AOC (Protected Designation of Origin) specifications. It is therefore essential to manage efficiently water supply in order to stick to the recommendations to both preserve water resource as well as yield and quality targets on vineyards. In addition, considerable thoughts are ongoing in regions that are increasingly suffering from drought such as those mentioned in the review [6] which describes solutions to improve water use efficiency. Proposed solutions include changes in agronomic practice (increasing soil water storage capacity, reducing direct soil water loss, using cover crops) and genetic improvements. Adaptive strategies [7] are also used to cope with environmental stress such as precision viticulture (use of decision models



or field captors) or in field technique (application of protective compounds). Multi-level strategies are also proposed to promote sustainable water use from the plant management to the consumer [8].

In this context, decision-support tools play a very important role in water status monitoring for vineyards. These tools allow following the evolution of the vine water status through the season and focus water supply on some areas or plots which suffer the most of water stress. They also could legitimize authorization requests for water supply but also provide proof of reasonable use of irrigation. Furthermore, a spatialized information could be used to adapt some cultural practices and to map the evolution between different dates.

In order to do that, remote sensing could be advantageous in particular with the availability of Sentinel-2 (S2) constellation data [9]. Its free availability combined with its revisit frequency of 5 days are well suited for monitoring vineyard during all development stages, making it a powerful tool for a low cost service [10]. The other advantage of the S2 sensor is its spectral resolution with 3 bands in the visible domains (Blue, Green and Red), 3 bands in the Red-Edge (RE), 2 in the Near-Infrared (NIR) and 2 in the Shortwave Infrared (SWIR). This is a benefit compared to other satellites as those domains have the potential to provide critical insights on vegetation behaviour [10].

Various studies have demonstrated the capacity of S2 images to analyze vineyard vegetation. According to [11], S2 can be powerful to monitor vine development with a high temporal resolution and the utility of S2 spatial resolution have been proved by [12]. Moreover, an estimation of crop coefficient (Kc) was done with S2 time-series by [13] and another study already proved the feasibility of qualifying the impact of heatwaves on irrigated vineyard using S2 images ([14]). To our knowledge, only one recent study has evaluated the potential of S2 images to directly estimate vineyard water status [15]. This study highlighted good relationship between stem water potential (SWP) and vegetation indices (VI) using NIR and SWIR regions in 82 vineyard in 2017.

The SWIR domains are known to be related to water concentration in plants [16–19] as they correspond to the vibrations of the O–H stretch in water molecules [17]. The NIR domains also react to water status as they can be affected by leaf morphology and structure [20]. Another spectral domain often related to water content is the Red-Edge as it is related to leaf chlorophyll concentration [21–23], which can in turn be linked to the plant water concentration [24,25]. A previous study using hyperspectral field measurements has already validated these three spectral domains to be linked with vineyard stem water potential [26].

In order to monitor vegetation, VI are commonly used since decades starting with the Normalized Difference Vegetation Index (NDVI) [27]. To predict water status, a previous study relied on a relationship between NDVI and crop coefficients [28] but others studies highlighted VI with RE and SWIR positions to be more related to vine water status [15,29]. Besides, as it becomes easier to implement Artificial Intelligence algorithms, more and more studies focus on the use of machine learning that uses all available spectral bands. This allows to take into account all the information included in the spectra [30–32]. There are nevertheless still obstacles to their operational use, especially considering their integration into automatic processing chains already in place in companies for example.

This paper investigates the potential of using S2 images to predict and monitor vine water status throughout the growing season on a large scale with a high temporal resolution. The database used in this study is composed of 36 vine plots in Mediterranean regions in the south of France with several grape varieties and grass management over three years (2018, 2019, 2020). Stem water potentials allowing to estimate vine water status were measured in the field from pea-size to harvest and S2 L2A images were downloaded at the closest dates. The objective is to find the best machine learning algorithm and the best feature to use Vegetation Index or reflectance bands values to predict stem water potential values of Mediterranean vineyard monitored during 3 years.

## 2. Materials and Methods

Figure 1 presents the materials and methodology used in this study. The first step is the data acquisition including S2 images acquisition and vine water status measurements on vine subplots (Section 2.1). Then the data preprocessing (Section 2.2) consists of resampling and merging S2 images as well as average the SWP values at the subplot-level.

In order to predict the water status with S2 images, two feature types were extracted from the images. The first one is the reflectance values of the bands. The second feature type is a set of selected VIs that are known to be correlated with water status (see Section 2.3 for more details).

Finally, analyses were performed in two steps. The first step evaluates the performance of multiple supervised regression algorithms to predict the SWP using S2 bands or VIs (Section 2.4.3). The aim is to determine the best feature type (S2 bands values or VI) and the best algorithm to use in order to predict SWP values in terms of accuracy but also versatility for the whole dataset (the one that works best in most cases). The second step explores the best models (Section 2.4.4) in terms of features importance (which S2 bands are needed) and analyzes data distribution.

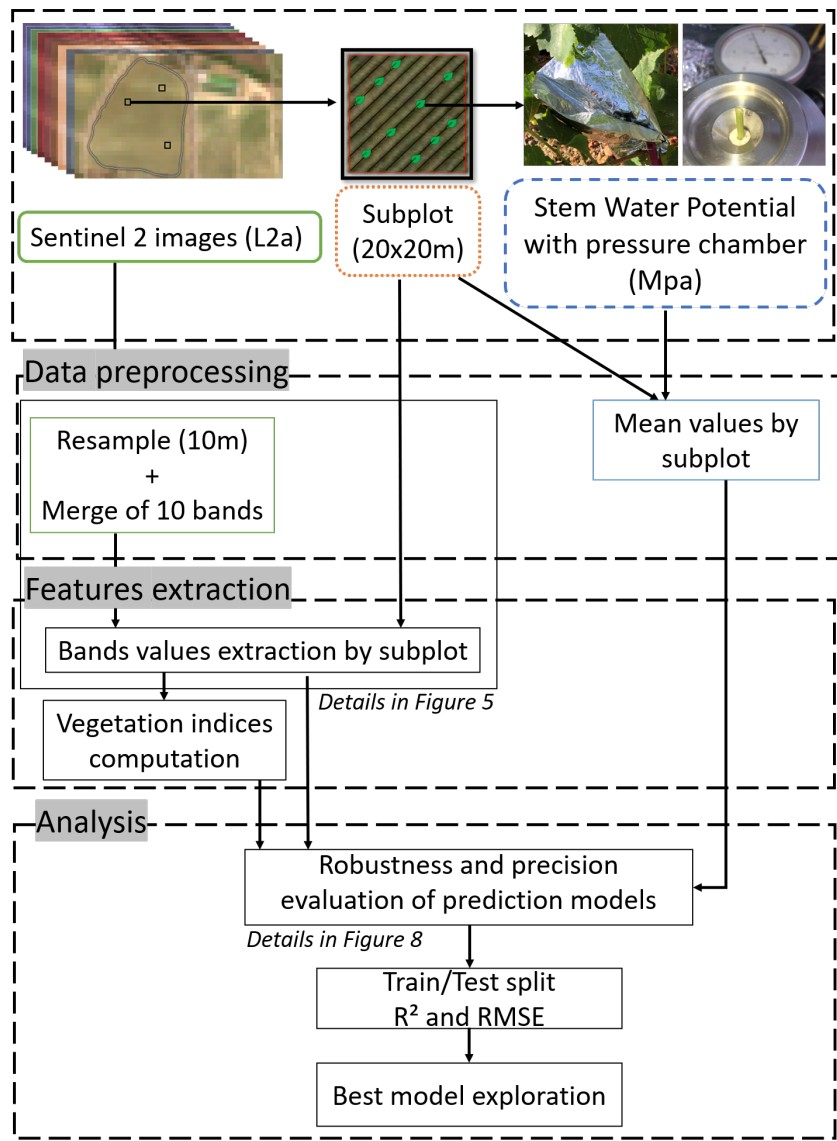

**Figure 1.** Flowchart of the method from the features extraction to the analysis step.

### 2.1. Data Acquisition

2.1.1. Study Sites

The study was conducted over three years with thirty six vine plots in the South of France (Figure 2). Most of the data were acquired on Syrah variety, one of the most common red grape varieties in the Mediterranean region. However, in order to complete the database, several other varieties were added during the study. In 2018, the measurements were focused on eleven vine plots with Syrah variety in the Minervois AOC area near Carcassonne. In 2019, five vine plots have been chosen with Syrah, Grenache and Chardonnay varieties near Carcassonne, Beziers and Montpellier. In 2020, twenty vine plots with six vine varieties were monitored near these previous city but also near Marseille.

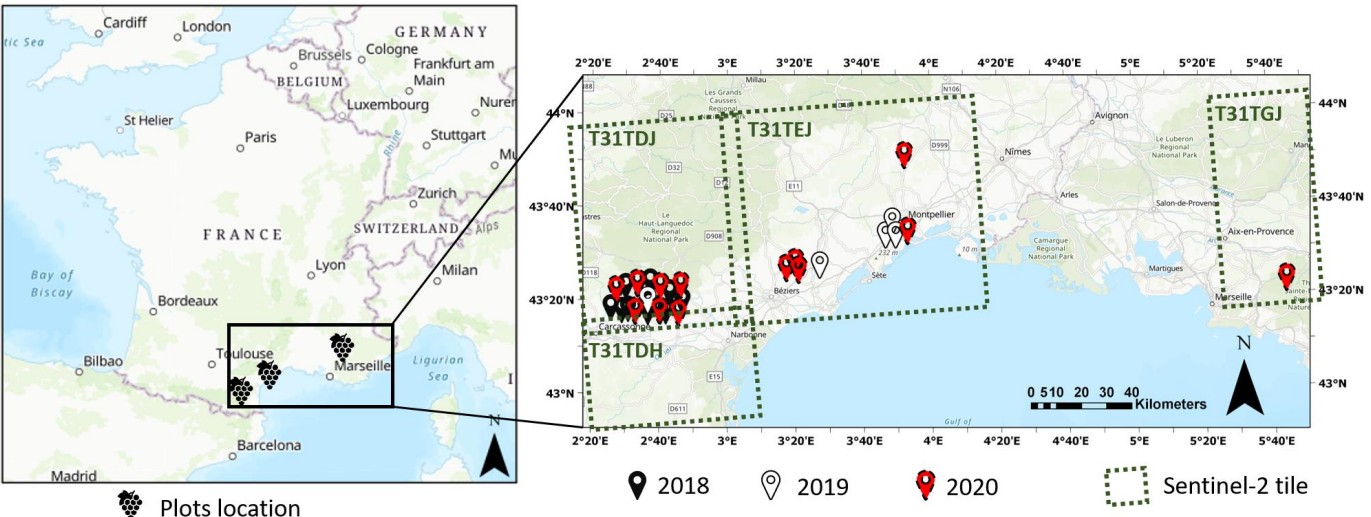

Sources: Esri, Airbus DS, USGS, NGA, NASA, CGIAR, N Robinson, NCEAS, NLS, OS, NMA, Geodatastyrelsen, Rijkswaterstaat, GSA, Geoland, FEMA, Intermap and the GIS user community

**Figure 2.** Location of the study vine plots for 2018, 2019 and 2020 and position of the corresponding S2 tiles.

Plots were all about 1 ha in area and in majority ungrassed (67%) and are all managed in bilateral cordon with an inter-row distance of 2 or 2.5 m and an inter-vine distance of 1 m. An example of three inter-row management is given in Figure 3. Particular attention has been paid to choose plots with no evident sign of diseases such as esca, Black dead arm or chlorosis.

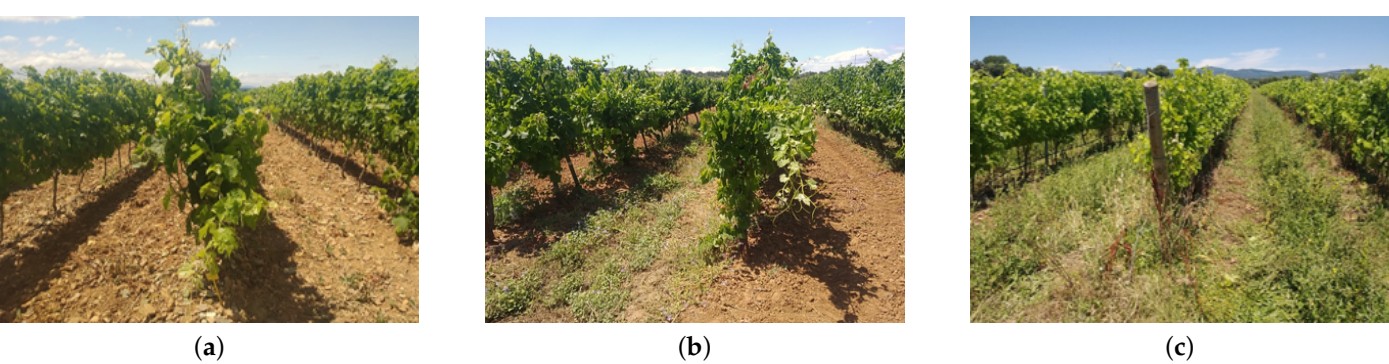

| (a) | (b) | (c) |

**Figure 3.** Inter-row management example. (**a**) Plot ungrassed. (**b**) Plot with grass one row on two. (**c**) Plot totally grassed.

Our study spans over three years with different types of vintage being represented. The year 2018 was a particularly exceptional year in terms of precipitation until June (2nd wettest year since 1975 with a cumulative precipitation of 971 mm) but with a heatwave event in the beginning of August [33]. The year 2019 was a very dry year (4th driest year since 1950) with a cumulative rainfall of 400 mm, i.e., 2 times less than in 2018. Additionally

average temperatures from July to October were above normal with a difference of up to 2 °C [33]. The year 2020 was similar to 2019, i.e., quite dry, but with a more humid spring [34]. Table 1 gives details for the months related to the vine development. Particularities are highlighted in red for a lack of water or warm temperatures, blue for too much water and cool temperatures, green for normal data and brown for contrasting rainfall or temperatures.

**Table 1.** Summary of the weather from May to September for 2018, 2019 and 2020 adapted from technical reports of Departmental Council of the Hérault region. Particularities are highlighted in red for a lack of water or warm temperatures, blue for too much water and cool temperatures, green for normal data and brown for contrasting rainfall or temperatures.

| Year | Month | Rainfall | Temperatures | Events |
|------|-------|----------|--------------|--------|
| 2018 | May | Very excessive | Alternating hot and very cold | Thunderstorms, hail and snow |
| | June | Very excessive | In accordance with season | |
| | July | No rainfall | Mild to warm | |
| | August | Normal | Very hot | Storms, heatwave the first week |
| | September | Very low | Very hot | |
| 2019 | May | Deficit | Cool to very cool | |
| | June | Deficit | Average to cool | Heatwave from the 26th to the 28th |
| | July | Deficit | Very hot | Heatwave, hail and fire |
| | August | Deficit | Hot to very hot | |
| | September | Constrasting | Warm to very warm | |
| 2020 | May | Heterogeneous | Mild to hot | |
| | June | Heterogeneous | Cool, warm the last week | |
| | July | Poor | Warm to hot | Hail and fires |
| | August | Heterogeneous | Warm | |
| | September | Heterogeneous | Warm | |

### 2.1.2. Experimental Design

In each vine plot, from one to six subplots of 20 × 20 m were defined for data acquisition. The number and location of the subplots were chosen according to three main criteria: (1) the S2 20 m pixel grid in order to have a good representation of the variability in the entire pixel, (2) plot soil and/or vegetation variability as reported by agronomic expert or vine-growers and (3) the time needed to perform the SWP measurements in the field (example in Figure 4). Vis for SWP measurements were selected in order to have a good representation of the entire pixel and according to the rows orientation inside the pixel.

Table 2 gives details about the number of plots and subplots for each year, by inter-row management and by variety. As expected, the Syrah variety is the most represented as it is the most common in the region but other grape varieties were added during the investigation in order to increase the variability within the database.

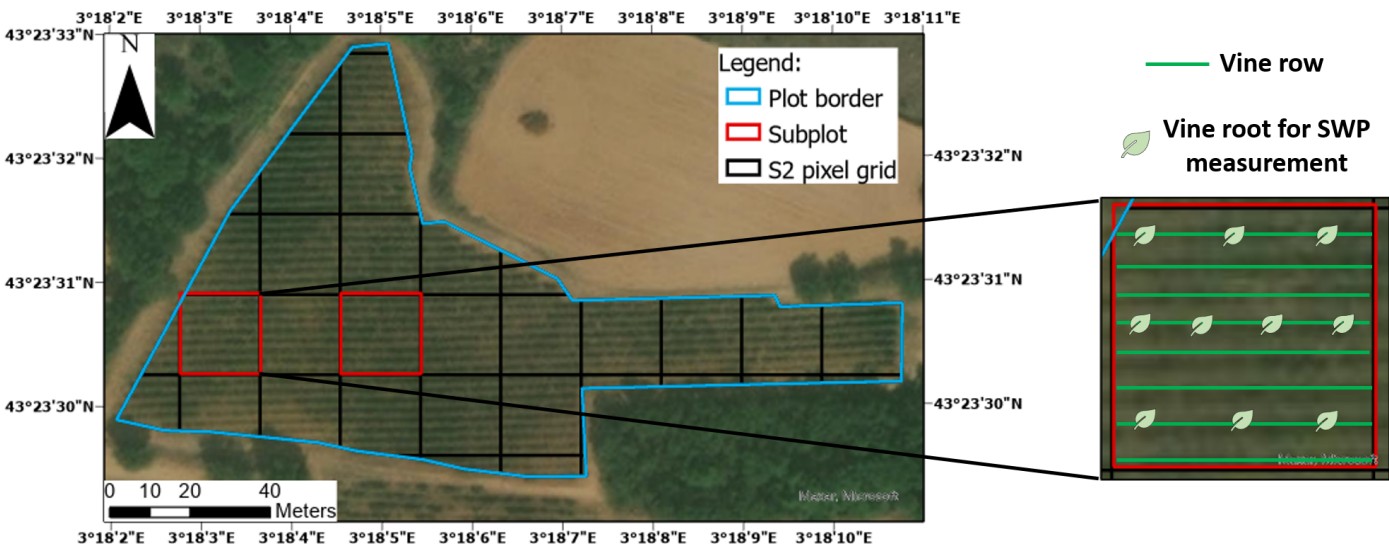

**Figure 4.** Example of 2 subplots in a plot and location of 10 SWP measurements in a subplot. The number and location of the subplots are chosen according to (1) the S2 20 m pixel grid in order to have a good representation of the variability in the entire pixel, (2) plot soil and/or vegetation variability as reported by agronomic expert or vine-growers and (3) the time needed to perform the SWP measurements in the field. Vis for SWP measurements were selected in order to have a good representation of the entire pixel and according to the rows orientation inside the pixel.

**Table 2.** Number of grass, ungrass and total plot and subplot for each year and their grape variety.

|  | 2018 | | 2019 | | 2020 | | Total | |
|---|---|---|---|---|---|---|---|---|
|  | **Plot** | **Subplot** | **Plot** | **Subplot** | **Plot** | **Subplot** | **Plot** | **Subplot** |
| Total | 11 | 18 | 5 | 32 | 20 | 53 | 36 | 103 |
| Inter-row management | | | | | | | | |
| Grass | 4 | 7 | 0 | 0 | 8 | 20 | 12 | 27 |
| No grass | 7 | 11 | 5 | 32 | 12 | 33 | 24 | 76 |
| Grape Variety | | | | | | | | |
| Syrah | 11 | 18 | 3 | 19 | 7 | 18 | 21 | 55 |
| Grenache | 0 | 0 | 1 | 6 | 1 | 2 | 2 | 8 |
| Chardonnay | 0 | 0 | 1 | 7 | 0 | 0 | 1 | 7 |
| Merlot | 0 | 0 | 0 | 0 | 1 | 3 | 1 | 3 |
| Cabernet Sauvignon | 0 | 0 | 0 | 0 | 2 | 5 | 2 | 5 |
| Caladoc | 0 | 0 | 0 | 0 | 1 | 6 | 1 | 6 |
| Mourvèdre | 0 | 0 | 0 | 0 | 1 | 1 | 1 | 1 |

### 2.1.3. Water Status

In order to assess the water status of the vine subplot, the stem water potential was measured on several vis (rSWP) as explained in [26]. The steps of the measurement are described below:

- Bagging mature leaves at 10 a.m. in order to close stomata and balance the sap pressure between plant and leaf;
- Removing leave with stem 3 or 4 h after, and set up quickly in the pressure chamber;
- Recording the pressure in MegaPascal (MPa) required to squeeze the first drop of sap out of the stem.

Between five and ten stems were measured (one stem by root) in each subplot as shown in Figure 4 which represents more than 3200 rSWP measurements in total for the three years. Measurements were carried out in maximum in 3 rows to optimize the time in

the field while trying to take into account the variability inside the subplot. Values range from 0 for water comfort to $-2$ MPa for extreme water stress.

### 2.1.4. Sentinel-2 Images

The S2 images were acquired over the studied vine plots and processed in L2A level [35,36] from the THEIA platform (theia.cnes.fr/, online accessed on the 6 January 2021). This platform was created at the end of 2012 by 9 French public institutions involved in earth observation and environmental sciences. This scientific and technical structure aims to facilitate the use of satellite images by making them available to the scientific community. It provides Sentinel-2 surface reflectances for each available date, corrected for atmospheric effects using the MAJA processing chain [37], and complemented with a cloud mask [38]. The images were chosen according to two criteria: (1) no cloud covering over the subplots and (2) a maximum of 5 days between the image date and the SWP measurement date. 80% of the samples had no more than 2 days apart between the field measurements and the date of the image. These multispectral optical images include 13 spectral bands in the visible, the near infra-red (NIR) and the shortwave-infrared (SWIR) spectral region [39]. 10 bands were used in this study with a spatial resolution of 10m and 20 m. The three other bands were not adapted for this study considering their spatial (60 m) and spectral resolution (in the atmospheric absorption domains for example). Details about the spectral and spatial resolutions of the bands used for the analysis are described in Table 3.

**Table 3.** Sentinel-2A spectral and spatial resolution of bands used in this study.

| Band | Spectral Region | Wavelength Range (nm) | Spatial Resolution |
|------|-----------------|-----------------------|--------------------|
| B2 | Blue | 458–523 | 10 m |
| B3 | Green | 543–578 | 10 m |
| B4 | Red | 650–680 | 10 m |
| B5 | Red-Edge 1 | 698–713 | 20 m |
| B6 | Red-Edge 2 | 733–748 | 20 m |
| B7 | Near Infrared | 779–793 | 20 m |
| B8 | Near Infrared | 785–899 | 10 m |
| B8a | Near Infrared | 855–875 | 20 m |
| B11 | Shortwave Infrared | 1565–1655 | 20 m |
| B12 | Shortwave Infrared | 2100–2280 | 20 m |

In total, 306 S2 images were available from June to September on the 4 concerned tiles (Figure 2) for 2018, 2019 and 2020. Taking into account our 2 criteria (no cloud and close to field measurements dates), 38 images were kept. Details of the dates of these images for each year and each tile are given in Table 4.

**Table 4.** Dates of the S2 images used for 2018, 2019 and 2020 for each S2 tile. Images were chosen among all the images available as there is no cloud over the subplots and the date is close to the in field measurements (Maximum difference of 5 days).

| S2 Tile | Year of Study | June | July | August | September |
|---------|---------------|------|------|--------|-----------|
| T31TDH/TDJ | 2018 | 27th and 30th | 30th | 1st | 8th and 10th |
| | 2019 | 17th | 5th, 17th and 25th | 14th | |
| | 2020 | 19th and 21st | 1st, 4th, 16th and 29th | 10th, 15th and 25th | |
| T31TEJ | 2019 | 17th and 22nd | 2nd and 17th | 16th and 26th | |
| | 2020 | 16th and 26th | 6th, 11th, 16th, 26th and 31st | 10th, 15th and 25th | |
| T31TGJ | 2020 | 28th | 2nd | | |

### 2.2. Sentinel-2 Images Preprocessing

The 20 m bands have been resampled to 10 m using the 8th band at 10 m resolution and then the ten bands have been merged in only one multispecral image. Figure 5 illustrates the data processing procedure.

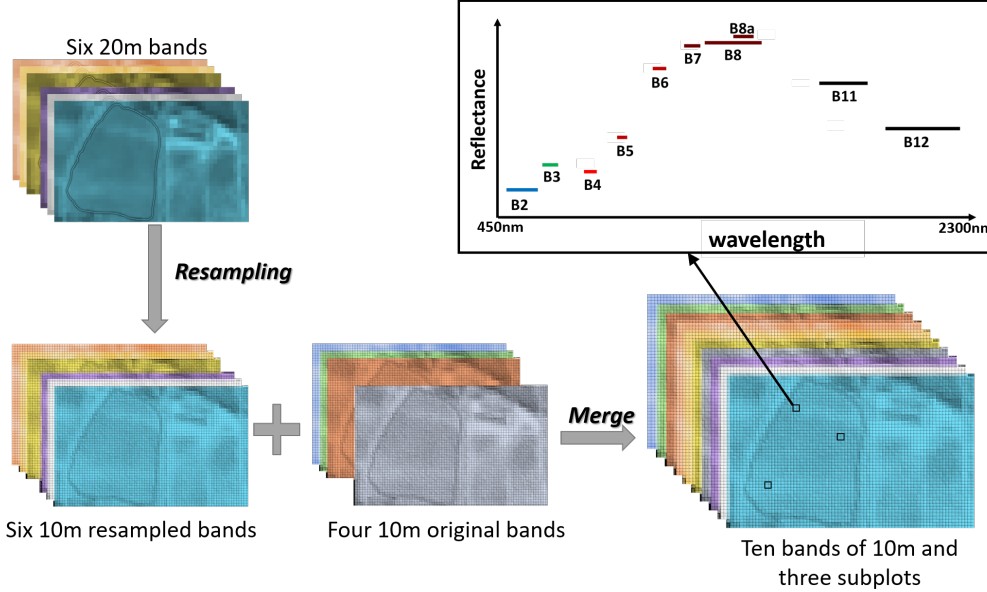

**Figure 5.** Flowchart of the resampling and the merge of S2 bands used in this study. Example for an image with only one vine plot with three subplots. This method as been used for the 38 images and the 103 subplot.

### 2.3. Features Extraction

2.3.1. Sentinel-2 Reflectance Bands Values

Once the ten S2 bands are merged together in one multispectral image, the values of each band are extracted for each subplot by averaging the value of the four 10 m pixels contained in the subplot. Pixels values of each band from THEIA platform have been divided by 10,000 in order to convert them to reflectance and to compute VI.

2.3.2. Vegetation Indices

Several VI have been chosen according to their ability to establish a link with the vine water status following a previous study [26]. The NDVI was also evaluated as it is a common VI usually used to monitor the vegetation. The details of the seven VI and their calculations formula are given in Table 5.

**Table 5.** Multispectral vegetation indices.

| Index (Abbreviation) | Main Use | Formula Used with S2 Bands | Reference |
|---|---|---|---|
| Normalized Difference Vegetation Index (NDVI) | Vigor | $\frac{B_8-B_4}{B_8+B_4}$ | [27] |
| Normalized Difference Red-Edge (NDRE 1/2) | Chlorophyll, Water | $\frac{B_8-B_{5/6}}{B_8+B_{5/6}}$ | [40] |
| Inverted Red-Edge Chlorophyll Index (IRECI) | Chlorophyll | $\frac{B_8-B_4}{\frac{B_5}{B_6}}$ | [41] |
| Red-Edge Chlorophyll Absorption Index (RECAI) | Chlorophyll | $\frac{B_8-B_6}{B_3}*\frac{B_6}{B_3}$ | [42] |
| Normalized Difference Infrared Index (NDII) | Chlorophyll, Water | $\frac{B_8-B_{11}}{B_8+B_{11}}$ | [43] |
| Red-Edge Position (REP) | Chlorophyll | $\frac{(\frac{B_4+B_8}{2})-B_5}{B_6-B_5}$ | [21] |
| Moisture Stress Index (MSI) | Water | $\frac{B_{11}}{B_8}$ | [18] |

### 2.4. Analysis

2.4.1. SWP Mean by Subplot

The five to ten rSWP measurements are averaged by subplot in order to reflect the pixel representation. In total, the database is composed of 349 observations, 50 for 2018, 146 for 2019 and 153 for 2020. An observation represents a subplot at a given date with its average value of SWP measurements.

2.4.2. Algorithms Description

Multiple supervised regression algorithms were used in the analysis (Table 6). These algorithms aim at mapping a set of observations (S2 reflectance values/VIs) with a continuous variable (SWP values). To be more specific, the K-nearest Kneighbors, ExtraTrees, Support Vector, linear and Bayesian Ridge model were evaluated. Supervised algorithms based on neural networks were avoided due to the limited number of training samples.

All the analysis have been performed using ScikitLearn Python package [44].

**Table 6.** Details of the algorithms used.

| Name of the Algorithms | Type of the Algorithm | Reference |
|---|---|---|
| Kneighbors regressor | k-nearest neighbors | [45] |
| ExtraTrees regressor | Decision Tree | [46] |
| Support Vector Regressor | Support Vector Machine | [47] |
| Linear regression and BayesianRidge model | Linear Model | [48] |

Each algorithm was tested (1) for all the database with all samples (2018, 2019, 2020, grass and ungrass), (2) for each year with grassed and ungrassed plots and (3) for each year with only ungrassed plots. Figure 6.

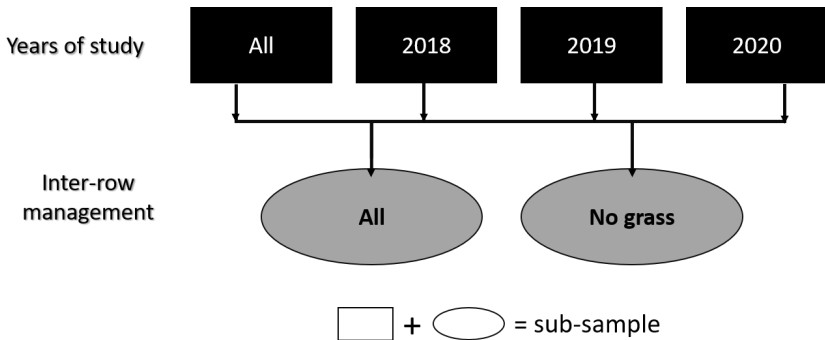

**Figure 6.** Illustration of the database sub-sample decomposition.

2.4.3. Robustness and Precision Evaluation of Algorithms

Figure 7 gives the analysis details. In a first part, a test of the robustness and precision of each algorithms is done with each feature (S2 bands values and VI) and each sub-sample (Years and inter-row management). A stratified suffle split is used to split the dataset into 10 parts (k = 10) in order to test the stability of the model. Into each of the 10 parts, attention has been paid to keep samples from all the studied development stages (a). For each 10 parts, 75% of the sample are used for the training and 25% for the test.

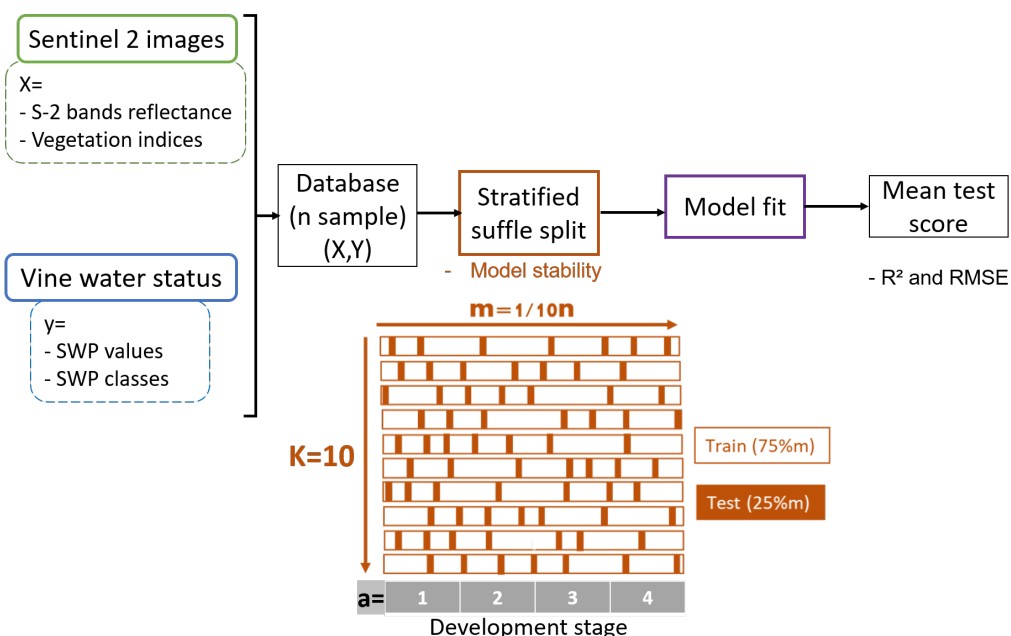

**Figure 7.** Flowchart details of analysis (n=349 for the entire dataset, 1: pea-size, 2: Pre-veraison, 3: veraison, 4: ripening).

- Regression score

For each split, the determination coefficient ($R^2$) and the Root Mean Square Error (RMSE) are computed for each algorithm according to the following formulas:

$$R^2 = 1 - \frac{\sum_{i=1}^n (y_i - \hat{y}_i)^2}{\sum_{i=1}^n (y_i - \bar{y})^2} \tag{1}$$

$$RMSE = \sqrt{\frac{\sum_{i=1}^n (\hat{y}_i - y_i)^2}{n}} \tag{2}$$

where $y_i$ and $\hat{y}_i$ are the $i$th measurement and corresponding predicted value for i = {1, ..., $n$}, $n$ is the number of measurements. and $\bar{y}$ is the mean of all the values. Note that the $R^2$ and RMSE are then averaged over the 10 splits.

In order to determine the best algorithm to use and the best feature type (S2 bands or VI), in addition to the model score, the best model is determined according to one more criterion: the versatility for all sub-samples (Years and inter-row management).

The results are shown for an RMSE > 0.3 (which corresponds to an average error of 30% for values ranging from 0 to −2 MPa).

### 2.4.4. Best Model Exploration

- Bands importance

It is important to highlight which bands of S2 are the most used in the algorithms to know which ones are the most interesting to predict vine water status.

The significance of the bands used to build the model is checked with the *p*-value and their coefficient in the model. Then, we recursively remove non-significant bands or bands with the smallest coefficients until scores ($R^2$ and RMSE) decline.

- Impact of experimental conditions on the result

The distribution of the values is verified with a single split of the data (75% for the training, 25% for the test) and the impact of inter-row management, years or study and variety are then examined.

## 3. Results

### 3.1. SWP Values

A box plot of the SWP values for each development stage is given in Figure 8. As expected, the mean values decrease during the season but differences are also observable for a same stage (from −0.4 MPa to −1.4 MPa at veraison for example).

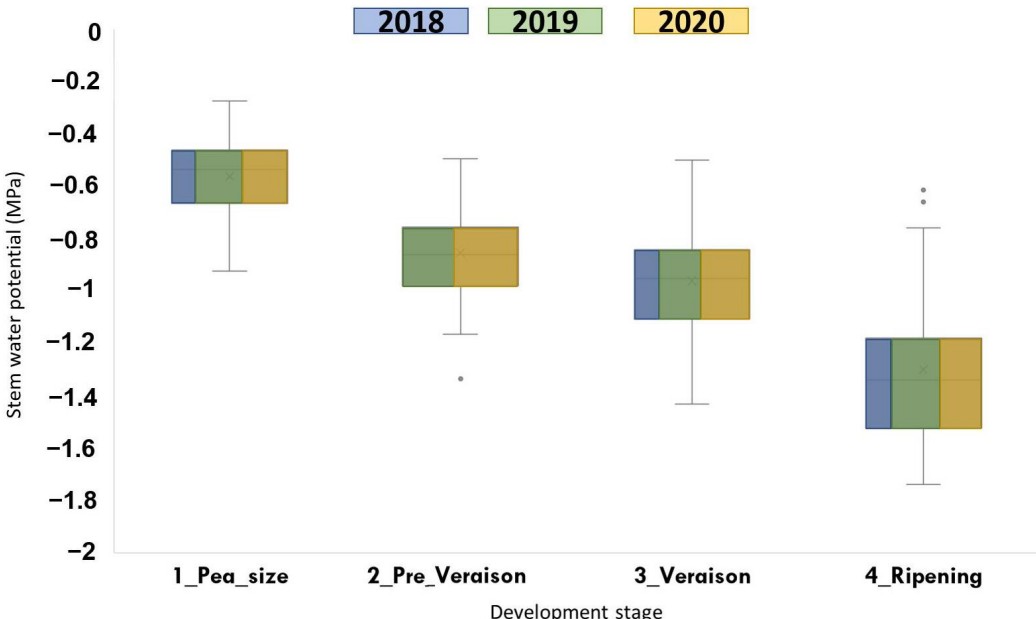

**Figure 8.** Box plot of SWP values (in MegaPascal) measured in field for each development stage covered by our study. The size of the colored rectangles indicates the percentage of data per year.

The detail for all statistics is given in Table 7 for each development stage of each year. The number of observations by development stage ranges from 15 to 49 according to the number of subplots, and the number of people available to do the measurements. The standard deviation ranges from 0.09 to 0.28 which indicates some significant variability in the measurements made in the same year at the same stage.

**Table 7.** Statistic description of SWP values for each development stage of each year (STD: standard deviation).

| Years | 2018 | | | 2019 | | | | 2020 | | | |
|---|---|---|---|---|---|---|---|---|---|---|---|
| **Development Stage** | 1 | 3 | 4 | 1 | 2 | 3 | 4 | 1 | 2 | 3 | 4 |
| Nb_observation | 15 | 17 | 18 | 26 | 39 | 32 | 49 | 27 | 48 | 37 | 41 |
| Min | −0.68 | −1.27 | −1.74 | −0.93 | −1.17 | −1.15 | −1.66 | −0.91 | −1.34 | −1.43 | −1.66 |
| Max | −0.33 | −0.50 | −0.60 | −0.28 | −0.53 | −0.54 | −0.62 | −0.42 | −0.49 | −0.72 | −0.80 |
| Median | −0.46 | −0.91 | −1.20 | −0.52 | −0.92 | −0.92 | −1.41 | −0.63 | −0.81 | −1.09 | −1.32 |
| Mean | −0.47 | −0.87 | −1.14 | −0.55 | −0.91 | −0.89 | −1.34 | −0.63 | −0.81 | −1.06 | −1.31 |
| STD | 0.09 | 0.21 | 0.28 | 0.18 | 0.16 | 0.16 | 0.25 | 0.13 | 0.15 | 0.18 | 0.25 |

### 3.2. Evaluation of the Robustness and Precision

The mean $R^2$ and RMSE of the 10 splits for each algorithm are computed for each feature type (VIs and bands values) for each sub-sample. An example of results is shown in Table 8 for the whole dataset (all years and all inter-row management). In this example, the best result is obtained using all S2 bands values with Bayesian Ridge and Linear Regression ($R^2 = 0.40$ and $RMSE = 0.26$).

**Table 8.** Results of scores exploration for all data (all years, all inter-row management). Best scores are highlighted in bold.

| Features | Scores | Algorithms | | | | |
|---|---|---|---|---|---|---|
| | | K-Nearest Kneighbors | ExtraTrees | Support Vector | Linear Model | Bayesian Model |
| All S2 bands | R2 | 0.29 | 0.39 | 0.37 | **0.40** | **0.40** |
| | RMSE | 0.28 | 0.26 | 0.26 | **0.26** | **0.26** |
| NDVI | R2 | 0.03 | 0.03 | 0.03 | 0.04 | 0.04 |
| | RMSE | 0.34 | 0.39 | 0.33 | 0.33 | 0.33 |
| NDRE1 | R2 | 0.07 | 0.05 | 0.03 | 0.004 | 0.003 |
| | RMSE | 0.34 | 0.41 | 0.33 | 0.33 | 0.33 |
| NDRE2 | R2 | 0.14 | 0.06 | 0.02 | 0.04 | 0.04 |
| | RMSE | 0.35 | 0.42 | 0.33 | 0.33 | 0.33 |
| IRECI | R2 | 0.02 | 0.04 | 0.11 | 0.11 | 0.11 |
| | RMSE | 0.34 | 0.39 | 0.31 | 0.31 | 0.31 |
| RECAI | R2 | 0.02 | 0.04 | 0.02 | 0.02 | 0.01 |
| | RMSE | 0.36 | 0.33 | 0.34 | 0.34 | 0.33 |
| NDII | R2 | 0.04 | 0.05 | 0.07 | 0.09 | 0.09 |
| | RMSE | 0.34 | 0.41 | 0.32 | 0.32 | 0.32 |
| REP | R2 | 0.02 | 0.03 | 0.15 | 0.19 | 0.17 |
| | RMSE | 0.33 | 0.39 | 0.31 | 0.30 | 0.30 |
| MSI | R2 | 0.09 | 0.06 | 0.07 | 0.10 | 0.10 |
| | RMSE | 0.34 | 0.41 | 0.31 | 0.31 | 0.31 |

Table 9 summarizes all the best algorithms (best $R^2$ and RMSE) for each sub-sample (years and inter-row management). Only results with an $R^2 \geq 0.25$ and RMSE $\leq 0.3$ have been reported, the only vegetation index meeting these conditions is REP (with Red, NIR and Red-Edge bands) and only for 2019. Using all S2 bands reflectance always gives a better result than this VI. The linear and Bayesian model appear to be the most accurate models in most cases (4 times on 7). The results seem better with only the plot without grass cover ($R^2$ up to 0.48 for all plots in 2020 and to 0.58 for plots without grass in 2019). The result per year is often higher for $R^2$ than with all years aggregated but the RMSE does not decrease so much and never falls below 0.2.

The best algorithm that will be explored next is the linear regression since it appeared to be the best in all cases with the whole dataset (all years with or without grass).

**Table 9.** Summary of the best algorithms for every subplot with $R^2 \geq 0.25$ and RMSE $\leq 0.3$.

| Year | Inter-Row Management | REP | All S2 Bands |
|---|---|---|---|
| All | All | $R^2 < 0.25$ | $R^2 = 0.40$ RMSE = 0.26 Bayesian Ridge model and Linear model |
| | No grass | $R^2 < 0.25$ | $R^2 = 0.48$ RMSE = 0.24 Linear model |
| 2018 | All | $R^2 < 0.25$ | $R^2 < 0.25$ |
| | No grass | $R^2 < 0.25$ | $R^2 = 0.52$ RMSE = 0.2 Extra Tree regressor |

**Table 9.** *Cont.*

| | | | |
|---|---|---|---|
| 2019 | No grass | $R^2 = 0.27$<br>RMSE = 0.29<br>Bayesian Ridge model | $R^2 = 0.58$<br>RMSE = 0.22<br>Linear model |
| 2020 | All | $R^2 < 0.25$ | $R^2 = 0.48$<br>RMSE = 0.21<br>Bayesian Ridge model<br>and Linear model |
| | No grass | $R^2 < 0.25$ | $R^2 = 0.56$<br>RMSE = 0.21<br>Bayesian Ridge Model |

For the linear model with the complete dataset, the values of $R^2$ and RMSE seem steady over the 10 splits as $R^2$ ranges from 0.43 to 0.55 and RMSE ranges from 0.22 to 0.28 (Figure 9).

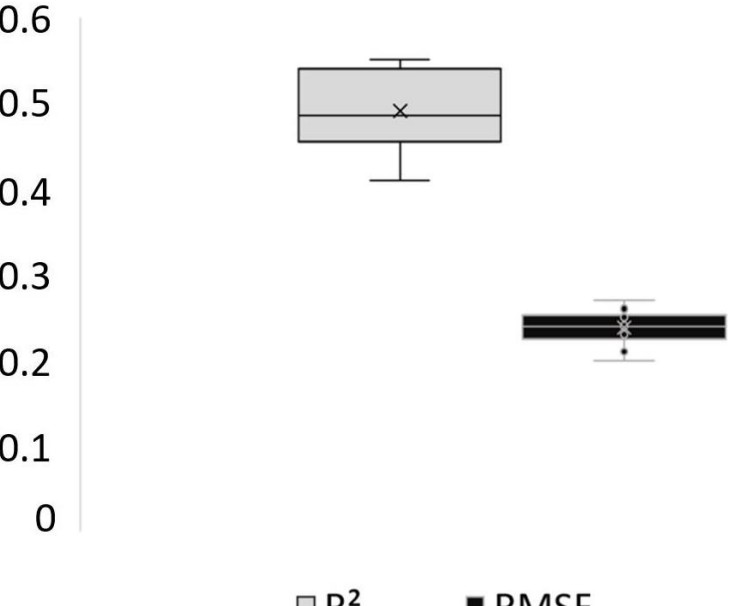

**Figure 9.** Boxplot of $R^2$ and RMSE over the 10 split for Linear model with the whole dataset (all years and all inter-row management).

*3.3. Best Models Exploration*

3.3.1. Bands Importance

According to their p-values (<0.05), all S2 bands were significant. In order to test if all the bands are really necessary to give a good result, we progressively removed the band with the smallest coefficient until an impact on the $R^2$ and RMSE scores is observed (Section 2.4.4). The result shown in Figure 10 highlights that only 4 bands are needed to perform as well as with 12 bands. Those 4 required bands are B4, B6, B8a and B12 in the Red, Red-Edge, NIR and SWIR regions, respectively.

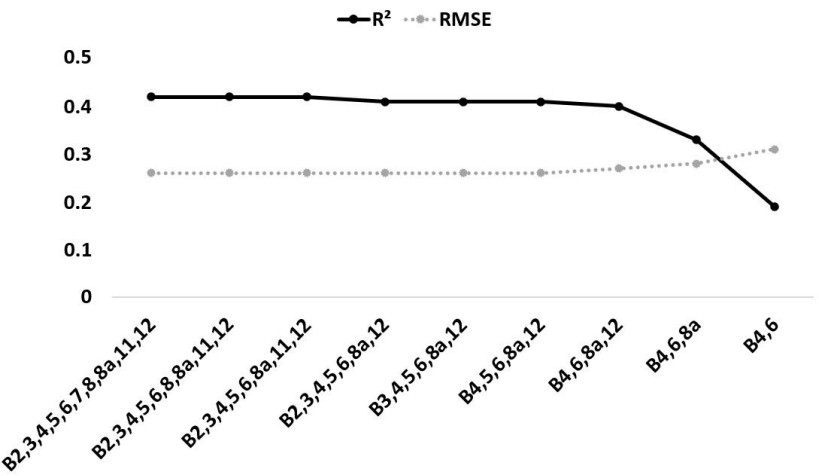

**Figure 10.** Best band to use for regression model.

### 3.3.2. Data Distribution and Impact of Experimental Conditions

This section aims at exploring the data distribution and the impact on the predictions when considering three experimental conditions: 1- Grass cover management, 2- Year of study, 3- Grapes variety. Comparison between observed and predicted SWP were plotted to test for discrepancies between reality and model's result obtained using the Linear Model based on the 4 bands of S2 highlighted in Section 3.3.1 (B4, B6, B8a and B12).

As a primary analysis, it can be noticed in Figure 11 that predicted SWP are a little overestimated until −1.4 MPa and a threshold effect seems to appear after.

- Inter-row management

Figure 11 shows the distribution of the predicted and observed SWP depending on the presence of grass in the inter-row. The scatterplot with and without grass superimpose pretty well.

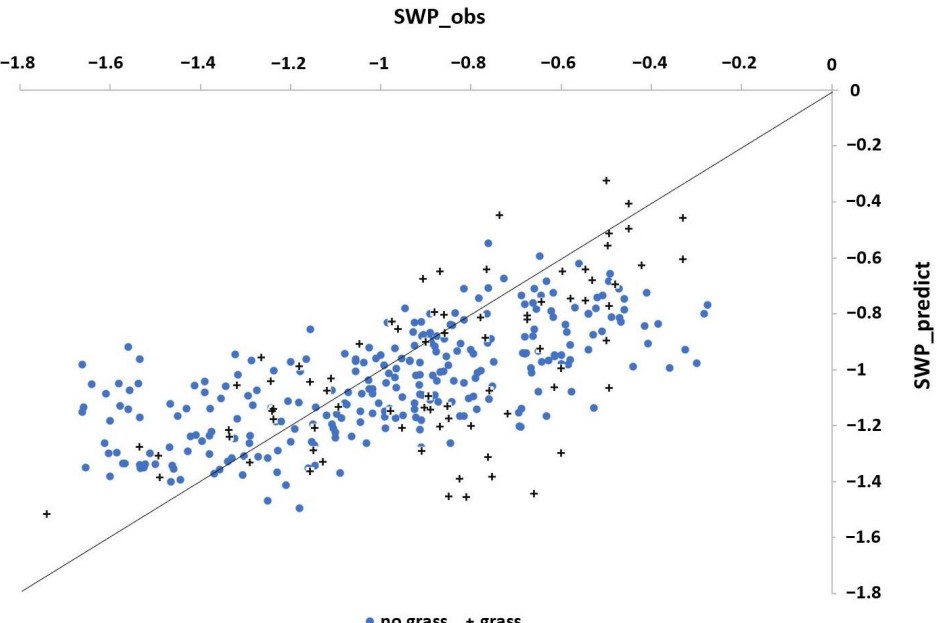

**Figure 11.** Distribution of predicted and observed data according to inter-row management.

- Grape variety

Figure 12 gives the data distribution according to grapes varieties. Even if the number of observations per grape variety is not the same Table 2, particular attention was paid

during the protocol so that measurements cover the whole range of potential water status (i.e., with just 4 points for the Mourvèdre, SWP values rang from −0.7 MPa to −1 MPa). One can also mention that the scatterplot of all varieties superimpose pretty well.

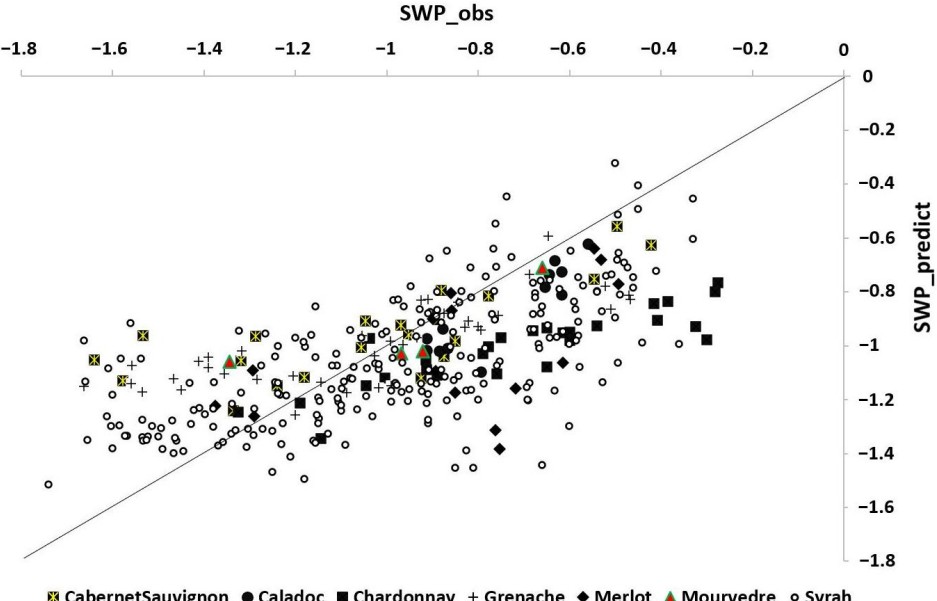

**Figure 12.** Data distribution according to grapes variety.

- Development stage

Figure 13 shows the distribution of predicted and observed SWP for each development stage. As expected, the relationship between development stage and SWP (i.e., increase of water stress (low SWP) as the season progresses from pea size to ripening) appears clearly. The second point, is the very large point cloud at the ripening stage with observed SWP spanning from −0.6 to −1.8. The prediction seems to be better for pre-veraison and veraison stage (data are closer to the straight line).

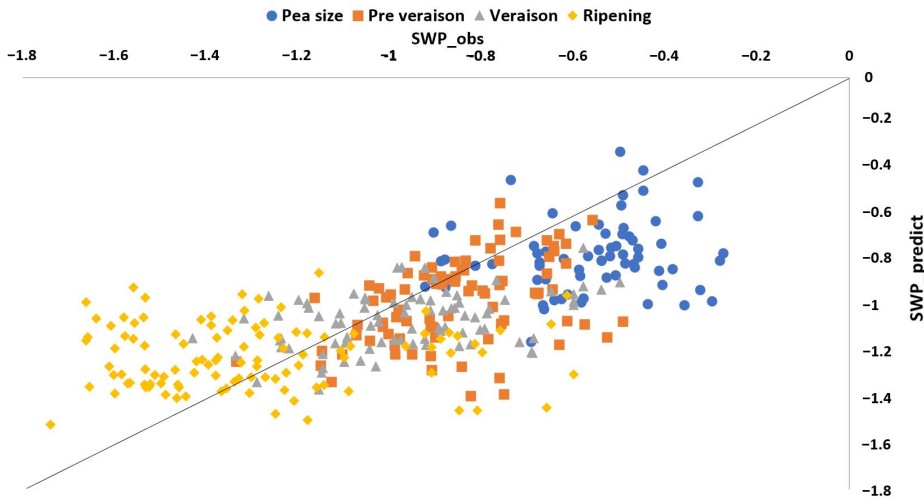

**Figure 13.** Data distribution according to development stage.

- Year of study

The observation of Figure 14 highlights the good distribution of the data for the 3 years with SWP values ranging from water comfort (−0.4 MPa) to water stress (−1.8 MPa) for each years.

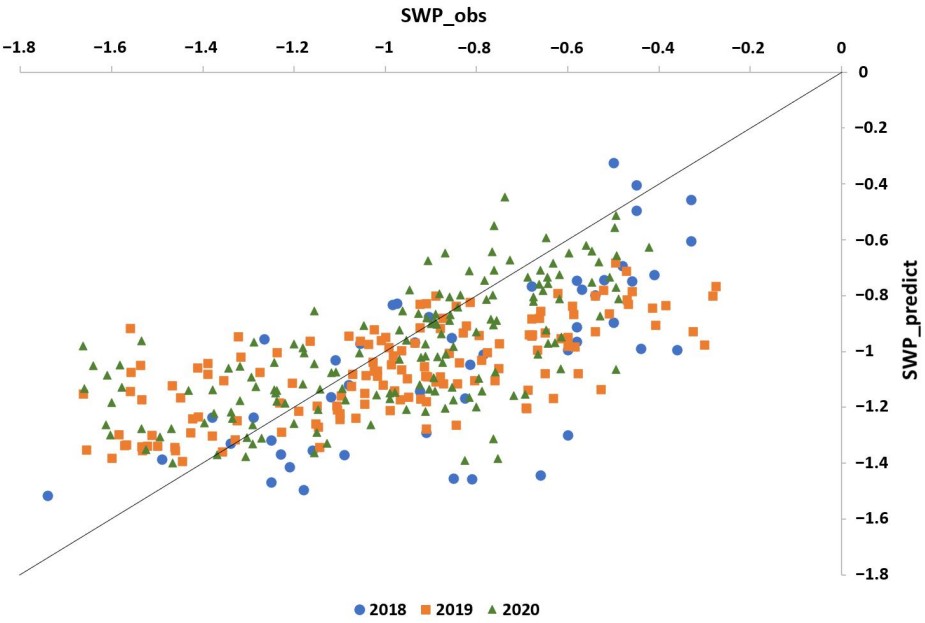

**Figure 14.** Data distribution according to study year.

*3.4. Comparison between NDVI, Best VI (REP) and Best Model with the Four S2 Bands*

In this study, both bands values from S2 images and VI were tested to predict vine SWP. According to the result, the commonly used NDVI (with B4 and B8) is not relevant at all (Figure 15a) and the only significant VI is the REP (with B4, B8, B5 and B6) but with a low $R^2$ of 0.19 for the whole dataset (Figure 15b). This VI gives less effective results than the model using the following S2 bands B4, B6, B8a and B12 (Figure 15c).

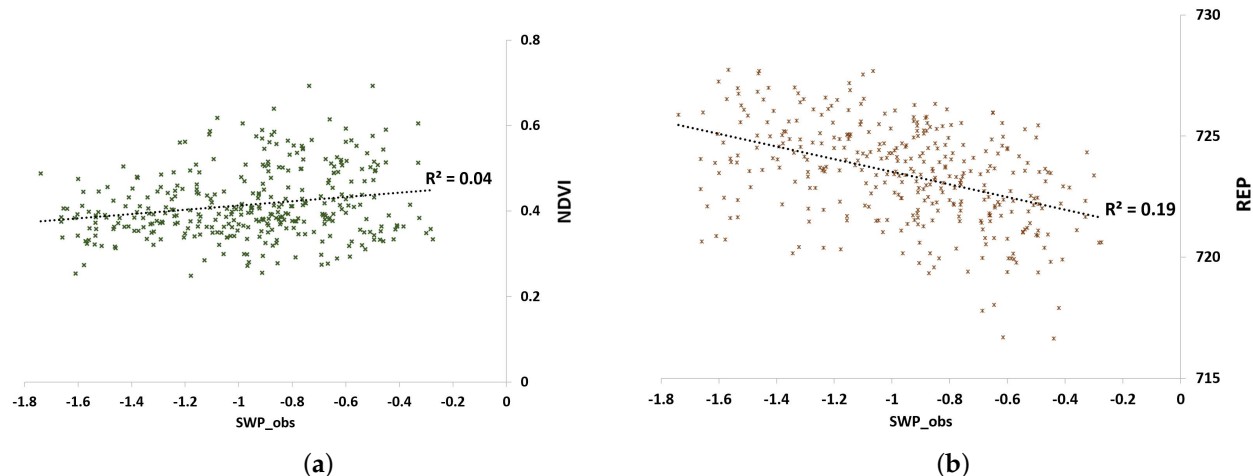

**Figure 15.** *Cont*.

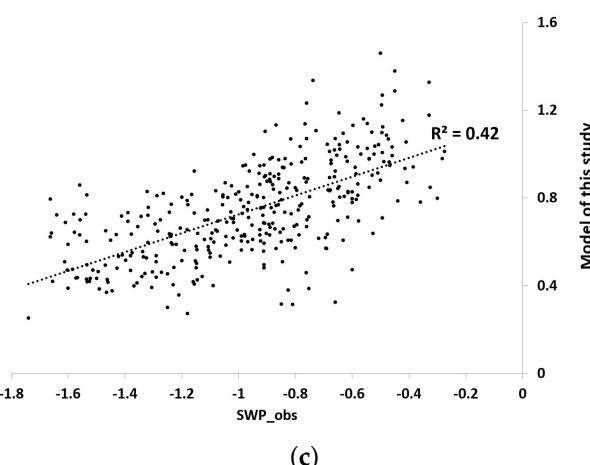

(**c**)

**Figure 15.** Distribution of the observed SWP according to (**a**) NDVI values, (**b**) REP values, (**c**) values of the model developed in this study.

## 4. Discussion

### 4.1. Significant Features

As previously reported in the literature, the most discriminant bands, highlighted in Section 3.3.1, are located in Red-Edge (B6) [49], NIR (B8a) [50] and SWIR (B12) [51]. The Red domain (B4) is also highlighted in this study as in [26]. This domain is related to pigments which also react to water stress (xanthophyll and chlorophyll) according to [52]. Except the REP, no other vegetation index appears to be interesting enough in our study to predict the vine water status with the algorithms we tested. Therefore the direct use of S2 bands seems to be more relevant.

### 4.2. Robustness of the Model

A first observation to highlight is that a threshold effect seems to appear for values to predict lower than −1.4 MPa. This can be explained by the fact that −1.4 MPa already reflects a very high water stress [53].

In the following paragraphs we discuss the impact of the inter-row grass management, as the diversity in grape varieties, the development stage and vintage, on the robustness of the model. Overall results show that in terms of prediction there is clearly a trend in the data distribution despite varying conditions. Specific effects are discussed in the 3 next paragraphs.

#### 4.2.1. Impact of Grass Cover

As shown in Table 9, the model performs better when only plots without grass in the inter-row are considered ($R^2$ = 0.48 and RMSE = 0.24 vs. $R^2$ = 0.40 and RMSE = 0.26). However, as can be seen in Figure 11, the results when also considering plots with grass show that only few points deviate from the trend. It can be explained by the fact that in Mediterranean region the vegetation in the inter-row is not always well developed especially in the middle of summer as shown by Figure 3. It would be interesting to extend the study with further investigation in the Bordelais or Burgundy regions where grass in the inter-row is more developed.

#### 4.2.2. Impact of Grape Variety

Most of the measurements were made on Syrah grape variety (Table 2) because it is the most common one in the study region. Nevertheless additional varieties were added in 2019 and 2020 to expend the scope of our initial survey. This way we were able to include both isohydric (Cabernet-Sauvignon, Grenache, Caladoc) and anisohydric grape varieties (Syrah, Chardonnay, Merlot, Mourvèdre) which behave differently to water stress.

Anisohydric varieties maintain their stomata opened to ensure photosynthesis and can be damaged in case of strong water stress while isohydric varieties close their stomata when the first signs of drought appeared and thus limit their photosynthesis [54]. Despite these distinct behaviors, we do not see any real differences in the predictions as shown in Figure 12 and the model seems to be able to predict correctly the SWP both for isohydric and anisohydric varieties. To extend the study it would be interesting to focus on more isohydric varieties and to add other varieties common in other regions (e.g., Merlot in the Bordelais region).

### 4.2.3. Impact of Development Stage

Following the trends observed in Figure 13, the model appears to be less efficient when considering data at ripening stage. This may be due to other physiological mechanisms that affect the plant when under high water stress, such as sugar accumulation in berries and modification in leaf area to fruit ratios [55], that could also affect the optical properties of the leaves. The objective of the study was to develop a model that could be used for images acquired at any stage of vine development. Nevertheless, tests were made without data from ripening stage to evaluate the impact on predictions. The models results were ultimately not better (max $R^2 = 0.30$), indicating that general model is more relevant, even if it is more accurate only when considering pre-veraison and veraison stage. This is rather positive those stages are the ones where winegrowers have to be careful on water stress as it can delay the good development of the berries.

### 4.2.4. Impact of Years

Despite differences in precipitations and temperatures (detailed at the end of Section 2.1.1), one can see in Figure 14 that no clear difference can be observed between the predictions for the three years covered by our study.

### *4.3. Considerations for a Future Operational Service*

The results of this study brings promising perspectives to develop a future operational service to monitor vineyard water status with Sentinel-2. However, when dealing with S2 images the question of spatial resolution is crucial and special attention will be needed to different points in order to be able to implement its use for a potential future service:

- Avoid the pixels at the edge which can cover both the plot and a path or a forest bordering it for example. In order to avoid its so-called "mixed" pixels, an advice would be to apply a buffer around the edge of the plot of at least 5 m to keep and interpret only the pixels fully included in the plot,
- Be careful with the size and shape of the plot in order to have a consistent number of pixels to interpret inside the plot,
- Consider the inter-row management and the soil management and/or the soil composition which can impact the observed signal since vineyard is conducted in row and the inter-row is also visible with 10 or 20 m pixel. Maybe think to use two models according to the grass cover management (one for ungrassed field and one for grassed) by improving the number of grassed vine field in the database.

In order to bring a new operational service it will be necessary to take into account these considerations but also to be able to add data over the years to improve the robustness of the model.

A next step is in progress with winegrowers in order to evaluate the spatial and temporal coherence of water status maps obtained with the model using the four S2 bands.

## 5. Conclusions

This study investigates the feasibility of using S2 images to monitor vine water status. The dataset is composed of 349 samples acquired during 3 years in the Mediterranean region. Water status in field measurements (SWP) and S2 images were acquired all along the vegetation cycle (from pea size to ripening) in 2018, 2019 and 2020. Our analyses allow

to highlight the best features to use from S2 images to predict SWP values: the four bands values in Red, Red-Edge, NIR and SWIR spectral domains. The regression linear model gives promising results to predict vine water status from different years, grape varieties and grass management. The model found will be more relevant for veraison stage than ripening. The characteristics of S2 make it possible to use this model to monitor vine water status at a large spatial scale (in a whole region for example) with 10m resolution and a little time of revisit (5 days). However, it is necessary to take into account the particularity of vine management in row and not to forget the potential impact of grass cover or soil management/composition to the result interpretation. The first winegrowers feedbacks are encouraging and further investigations will be done to increase the dataset and improve the model with other grape varieties and grass cover-management for example.

**Author Contributions:** Conceptualization, E.L.-P., H.C. and S.D.; methodology, E.L.-P., H.C., S.D. and J.R.; validation, H.C., S.D. and V.C.; formal analysis, E.L.-P. and M.A.; investigation, E.L.-P., S.D. and J.R.; resources, E.L.-P., H.C., S.D. and J.R.; data curation, E.L.-P., A.D.C. and M.A.; writing—original draft preparation, E.L.-P., H.C. and S.D.; writing—review and editing, E.L.-P., H.C., S.D., M.A. and A.D.C.; visualization, E.L.-P.; supervision, H.Clenet, S.D. and V.C.; project administration, E.L.-P.; funding acquisition, S.D. and H.C. All authors have read and agreed to the published version of the manuscript.

**Funding:** This research was funded by ANRT, TerraNIS, ICV and EI Purpan.

**Acknowledgments:** The authors thank all the trainees of ICV and TerraNIS involved in field measurements acquisition. They also thank winegrowers for allowing the realization of measurements in their fields and the Minervois Union in particular Marie Vidal-Vigneron.

**Conflicts of Interest:** The authors declare no conflict of interest.

## Abbreviations

The following abbreviations are used in this manuscript:

| | |
|---|---|
| MDPI | Multidisciplinary Digital Publishing Institute |
| DOAJ | Directory of open access journals |
| NIR | Near Infrared |
| NDVI | Normalized Difference Vegetation Index |
| S2 | Sentinel-2 |
| SWP | Stem Water Potential |
| SWIR | Short Wave Infra-Red |
| VI | Vegetation indices |
| REP | Red-Edge Position |
| RMSE | Root Mean Square Error |

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
