# Peer review of "Towards Vine Water Status Monitoring on a Large Scale Using Sentinel-2 Images"

_remotesensing, doi:10.3390/rs13091837_

Round 1
Reviewer 1 Report
Towards vine water status monitoring on a large scale using Sentinel-2 images
This study refers to the investigation of the potential use of Sentinel 2 images to predict and monitor vine water status throughout the growing season on a large scale with a high temporal resolution. The authors monitored thirty six plots that were located in South France for a period of three years (2018, 2019 and 2020). In each vine plot, from one to six subplots were defined for data acquisition and the stem water potential was measured on several vine roots (rSWP). The authors assessed four different machine learning algorithms on predicting vine water status through the Sentinel 2 imagery and compared them with 7 different vegetation indices. The authors concluded that the regression linear model gives promising results to predict vine water status from different years, grape varieties and grass management. However, it is necessary to take into account the particularity of vine management in row and not to forget the potential impact of grass cover or soil management/composition to the result interpretation.
Comments
Abstract
L7-9. I recommend the authors to use past tense.
L10. I suggest the authors to write the exact number of plots that they monitored.
L13-14. I recommend the authors to use past tense.
L15-16. I recommend the authors to rephrase the sentence and use past tense.
I suggest the authors to introduce some numeric results in the abstract (e.g. R2 of the best fitted model).
Introduction
I suggest the authors to provide a paragraph on the use of machine learning algorithms in viticultural studies in order to provide a link with the methods that they applied in their work. Also, information of vegetation indices is missing from the introduction. Additionally, the authors could highlight the benefits that the combination of imagery data derived from all satellite bands with machine learning algorithms, can offer when compared to vegetation indices.
Materials and Methods
I recommend the authors to use past tense. Also, the subsections Experimental design and Water status should be moved before the Sentinel-2 images subsection for helping readers to understand better the conducted study. The subsection SWP mean by subplot should be moved to results section with descriptive statistics.
Results
I recommend the authors to introduce tables with all the results on determination coefficients and the RMSE from the vegetation indices, bands and the algorithms before presenting Table 6.
Discussion
Figure 15 and the first paragraph of Significant features subsection should move to results section.
I recommend the authors to discuss for all the results presented in this study and not only for results related with the best fitted models.
The weather information during the study period, should be presented in the materials and methods section.
General Comment
I recommend this manuscript to be accepted after major revision. The study provides a method for utilizing satellite data through machine learning approaches for predicting the stem water potential in vines. Currently, there is a huge interest on new methods for utilizing information from large datasets such as the ones provided by satellites. However, the authors need to present better their results for supporting their work.
Reviewer 2 Report
I enjoyed reading this manuscript unveiling the utility of multispectral Sentinel 2 data for vine water stress in Southern France. The outcome of the study could play a significant role in understanding vine yards and their implications in the region. My comments and suggestion are:
- L3: “resistant varieties” could be replaced with “resilient varieties”
- The abstract should be revised, currently is focusing on the problem and background. It should briefly explain methods, quantitative results, and policy implementation.
- Introduction
Very good, the authors demonstrate a thorough knowledge of the published literature and highlight the importance and background to carry out this investigation. - Objectives could be rephrased, especially the last sentence 75-77 must be rephrased. And the author could avoid using unexplained acronyms.
Methods
The data sets used in the study and the methods are technically strong and well explained.
- Authors may consider explaining the datasets used in the study.
- The flow chart could include the image level obtained for the study (L2A?), dimension of subplots should be mentioned in the flowchart.
- VI could be calculated before pixel value extraction which could help later for implementation of the developed models.
- Metric to determine the vine yard water status could be mentioned.
- SWP should be written in full form or figure caption. Methods for finding the best model should be indicated.
- I could not find the dimensions of the plots and subplots in the section and what measurements were taken during the field surveys.
- Explain THIEA platform.
- How did the differences among the number of images in different months and sites affect the modeling and mapping?
- Authors could have considered a standard number of images for all the years and sites, just a thought.
- If the 20m grid was used then why the data was resampled to 10m?
- Figure 12 and Figure 13 indicate that a non-linear fit could be a better choice.
- Line 100 what authors mean by “ to complete the database”?
- Conclusion: No comments
Round 2
Reviewer 1 Report
The authors have properly addressed all comments. I recommend to accept the manuscript at its present form.
Reviewer 2 Report
I am fine with the revised version, the authors have responded to all the comments.